# Enhanced Symbiotic Characteristics in Bacterial Genomes with the Disruption of rRNA Operon

**DOI:** 10.3390/biology9120440

**Published:** 2020-12-03

**Authors:** Hyeonju Ahn, Donghyeok Seol, Seoae Cho, Heebal Kim, Woori Kwak

**Affiliations:** 1Department of Agricultural Biotechnology, Seoul National University, 1 Gwanak-ro, Gwanak-gu, Seoul 08826, Korea; poroko@snu.ac.kr (H.A.); sdh1621@snu.ac.kr (D.S.); heebal@snu.ac.kr (H.K.); 2eGnome, 26 Beobwon-ro 9-gil, Songpa-gu, Seoul 05836, Korea; seoae@egnome.co.kr

**Keywords:** bacterial regulatory system, symbionts, rRNA operon, transcription

## Abstract

**Simple Summary:**

Exploring the genomic changes that organisms have undergone to adapt to their specific environment is one of the most important processes in ecology and evolutionary biology. Here, we found that almost all rRNA operon-unlinked bacteria are symbiotic bacteria, which could be evidence of specific selective pressures in symbionts like genome reduction. This is meaningful and suggests that not only does the copy number variation of the rRNA operon sensitively respond to the bacterial lifestyle, but structural modification can also strongly reflect adaptation to the surrounding environmental conditions.

**Abstract:**

Ribosomal RNA is an indispensable molecule in living organisms that plays an essential role in protein synthesis. Especially in bacteria, 16S, 23S, and 5S rRNAs are usually co-transcribed as operons. Despite the positive effects of rRNA co-transcription on growth and reproduction rate, a recent study revealed that bacteria with unlinked rRNA operons are more widespread than expected. However, it is still unclear why the rRNA operon is broken. Here, we explored rRNA operon linkage status in 15,898 bacterial genomes and investigated whether they have common features or lifestyles; 574 genomes were found to have unlinked rRNA operons and tended to be phylogenetically conserved. Most of them were symbionts and showed enhanced symbiotic genomic features such as reduced genome size and high adenine–thymine (AT) content. In an eggNOG-mapper analysis, they were also found to have significantly fewer genes than rRNA operon-linked bacteria in the “transcription” and “energy production and conversion in metabolism” categories. These genomes also tend to decrease RNases related to the synthesis of ribosomes and tRNA processing. Based on these results, the disruption of the rRNA operon seems to be one of the tendencies associated with the characteristics of bacteria requiring a low dynamic range.

## 1. Introduction

Bacterial ribosomal RNA varies in number and typically forms an operon that comprises 16S, 23S, and 5S rRNA with an intergenic transcription spacer (ITS) [1,2]. The bacterial rRNA operon is co-transcribed to compose a ribosome for protein synthesis. Because of the importance of the rRNA operon, the distribution and properties of the rRNA operon related to the bacterial lifestyle have been studied repeatedly in analyses of diverse bacterial genomes. The rRNA operon copy number reflects the ecological adaptation of bacteria in response to available resources [3] and has been shown to decrease in bacterial communities during ecological microbial succession [4]. Depending on a doubling of the rRNA copy number, the maximum reproductive rate of bacteria also doubled and the efficiency of carbon use had a negative correlation to maximal growth rate and rRNA copy number [5]. Multiple 16S rRNA genes in the genome showed concerted evolution, where a sequence change must be eliminated or shared in every copy, which slows the evolution rate relative to single-copy genes [2].

The rRNA operon is co-transcribed into a single transcript like other bacterial operons and has been found to be under stringent regulation by multiple promoters, as studied in *Escherichia coli* [6,7]. The typical linked form of rRNA operon has been reported and studied in many bacteria; however, the unlinked rRNA operon has occasionally been reported from certain genomes. In *Pirellula marina*, the presence of unlinked rRNA operons was revealed from southern hybridization [8]. The distance between 16S and 23S rDNA of *Thermoplasma acidophilum* was at least 7.5 Kb, 6 Kb between 23S and 5S rDNA, and 1.5 Kb between 16S and 5S rDNA, which indicated an unlinked rRNA operon [9]. Moreover, *Buchnera aphidicola* had 16S rRNA operons including several ORFs but not 23S rRNA [10], and *Wolbachia pipientis* had a 23S-5S rDNA operon in one transcription unit, separately from the 16S rDNA in the other transcription unit [11]. A recent study on *Helicobacter pylori* showed independent maturation of the 16S and 23S-5S precursors separately transcribed from the unlinked rRNA operon, which is regulated by RNase III and the antisense RNA [12].

The extensive distribution of the unlinked rRNA operon has not been identified until recently. However, following on from the accumulation of microbial genomic sequence data, the latest research has identified rRNA operons’ disruption from genomic data and reported the spread of unlinked rRNA operons in diverse bacterial and archaeal taxa [13]. The research results first presented insights into the distributions of unlinked rRNA operons in slower-growing taxa living in soil and sediment with significantly fewer rRNA copies and RNase III genes. However, it remains uncertain what has influenced the disruption of the rRNA operon.

Therefore, we explored the genomic implications related to the unlinked rRNA operon using expanded and more rigorous curation data. Recently, issues such as assembly errors or incorrect taxonomic assignments have emerged for genomes registered in the public repository [14,15,16]. As these can affect our analysis, we used Genome Taxonomy Database (GTDB) [17], which reclassified bacterial taxonomy based on genome phylogeny and examined the consistency between annotation information and primer-based identification to use only reliable rRNA information. In addition, we determined whether the rRNA operon was unlinked based on the length of the internal transcribed spacer (ITS) and explored which bacteria have an unlinked rRNA operon and which environments they are closely associated with. Moreover, characteristics of bacterial genomes with the unlinked rRNA operon were analyzed in consideration of the association with their lifestyle. Because the operon is one of the microbial genetic regulatory systems, we believe that there will be a certain genetic selection associated with transcription/translation regulation or other functional metabolic pathways. Therefore, we investigated the functional gene alteration in rRNA operon-unlinked genome through the Clusters of Orthologous Genes (COG) annotation.

## 2. Materials and Methods

### 2.1. Bacterial rRNA Data Preparation

In this research, we referred to the NCBI RefSeq database (https://www.ncbi.nlm.nih.gov/refseq/) for information on 16S and 23S rRNA. We first downloaded the whole genome sequences (.fna) and gene annotation files (.gff) of 15,898 complete bacterial genomes (except genomes tagged “excluded_from_refseq” in the RefSeq assembly summary file) on 8 January 2020. From each genome annotation file, the information of 16S and 23S rRNA for the identification of rRNA operon was collected: chromosome IDs, whether circular or linear chromosome, whether sense or antisense rRNA, and start/end position. rRNAs marked “partial = true” in the annotation file were removed from the collecting process.

### 2.2. Taxonomy Reclassification

We used the GTDB (version release89) [17], which reclassified the taxonomy based on 120 ubiquitous single-copy proteins to correct misclassified taxonomy in NCBI and obtain a consistent taxonomy based on the genome phylogeny. Of the total 15,898 complete genomes, 10,008 genomes have already been reclassified in GTDB, and the remaining 5890 have been reclassified taxonomically by performing GTDB-Tk (version 0.3.3) [18]. Phylogenetic relationships were determined using pplacer [19], a dependency of GTDB-Tk, and visualized with iTOL [20].

### 2.3. Primer-Based Identification and rRNA Filtering

PCR products are generated in silico to remove a genome with misannotations of rRNA from NCBI and confirm whether it is an experimentally verifiable rRNA sequence. For targeting 16S rRNA gene, 27F: 5′-AGRGTTYGATYHTGGCTCAG-3′ and 1492R: 5′-TACCTTGTTAYGACTT-3′ [21] were used. In the case of the 27F, 5′-AGRGTT′Y′GATYMTGGCTCAG-3′ [22] and 5′-AGRGTTTGATY’H’TGGCTCAG-3′ [23] primers were combined. For 23S rRNA, 11a: 5′-GGAACTGAAACATCTAAGTA-3′ [24] and 2241r: 5′-ACCRCCCCAGTHAAACT-3′ [23] were used. We generated in silico products using EMBOSS-primersearch (version 6.6.0.0) [25] and filtered them by length (1000 bp < 16S rRNA product < 4000 bp, 1000 bp < 23S rRNA product < 6800 bp). The first time, up to two mismatches were set, and we compared them with the annotation information. Genomes with different rRNA profiles were permitted four mismatches and compared again. If there were still differences, the genome was not used for subsequent analyses.

### 2.4. rRNA Operon Identification

To identify the rRNA operon structure, the start/end position and direction of rRNA in RefSeq annotation was used. A schematic diagram summarizing the identification of the rRNA operon is given in Figure 1. As a first step, all rRNAs were arranged by the start site of the genome with sense/antisense information. Considering the direction of transcription, sense 16S rRNA and the following sense 23S rRNA were defined as rRNA operons. Likewise, antisense 23S rRNA and the following antisense 16S rRNA were defined as rRNA operons. The rRNAs that did not fit the operon structure described above were defined as single rRNA copies. Based on the ITS length, an rRNA operon with ITS over 1500 bp in length was designated as an “unlinked” rRNA operon [13] (Figure 2A). Depending on the presence or absence of an unlinked rRNA operon, genomes were classified into three groups: “linked”, “mixed”, or “unlinked” (Figure 2B‒D).

### 2.5. Correlation Analysis of Genomic Features of Symbiotic Bacteria

SymGenDB was used to identify symbiotic bacteria in Refseq complete genomes [26]. From the search for genomes related to Arthropoda, Mammalia, Nematoda, and Panarthropoda, species that had a NCBI taxonomy ID in the SymGenDB were classified as symbiotic bacteria. For species with two or more taxonomy IDs, if any of the IDs were in SymGenDB, they were classified as symbionts. The principal component analysis (PCA) and statistical tests on genomic features and the rRNA operon linkage status of symbiotic and nonsymbiotic bacteria were performed by R (version 3.4.4) [27].

### 2.6. Comparisons of Functional Genes between rRNA Operon-Linked and Unlinked Bacteria

To examine the functional differences between linked and unlinked genomes, the following two analyses were performed using the protein sequence (.faa file). First, proteins were classified into functional categories using eggNOG-mapper (version 2.0.1) with DIAMOND as the mapping mode [28,29]. This was performed at the phylum level with Spirochaetota (28 linked; 76 unlinked), Campylobacterota (308 linked; 191 unlinked), and Chloroflexota (11 linked; 27 unlinked), where the difference between the number of linked and unlinked genomes was not tripled. Second, we investigated how many RNases exist in the genome. The presence or absence of seven types of RNases registered in Pfams was determined using HMMER (version 3.3) with sequence search (HMMSearch) [30]. To reduce false positives, we used gathering cutoff with the “-cut_ga” option. The correlations of RNase number to rRNA operon-unlinked taxa were calculated using gls function in the nlme R package with the phylogenetic covariance model of the Brownian motion [31,32].

## 3. Results

### 3.1. Confirmation of rRNA Operon Status

In this study, we performed taxa reclassification from 15,898 RefSeq complete genomes and investigated the status of the rRNA operon in two ways. According to GTDB Statistics (https://gtdb.ecogenomic.org/stats, accessed 22 May 2020), around 45% of the taxonomy of 143,566 genomes was changed at the species level. We additionally performed GTDB-Tk on 5890 genomes not included in GTDB but in RefSeq. As a result of comparing GTDB reclassified taxonomy and NCBI taxonomy for 15,898 RefSeq genomes, 28% of genomes were changed at the species level. These included unclassified cases (4.8%) and those classified as a placeholder taxon (5.1%) because the scientific name was not yet available (Figure 3A). Reclassification was also shown at the higher rank with 7.4% (1174 genomes) at the phylum, 21.5% (3423 genomes) at the class, 27.3% (4334 genomes) at the order, 22.4% (3560 genomes) at the family, and 7% (1120 genomes) at the genus level. According to a recent proposal [33], “‒ota” is suffixed at the phylum level in the GTDB results. Therefore, we performed a comparison after removing the suffix at the phylum level.

Annotation data in RefSeq were defined by using the NCBI prokaryotic genome annotation pipeline (PGAP), which combines alignment-based methods with direct gene prediction from a sequence [34]. On the other hand, primer-based rRNA sequence identification targets universally conserved regions and selects those formed in a specific length range. Therefore, if these two identification methods are cross-validated, we can solve misannotation and experimentally unverifiable problems that can be caused by misassembly, variation in the rRNA sequence (possibly dropped by PGAP coverage filtering), primer binding site variation, or unknown sequence modification that disturbs identification mechanisms. From our comparison, there were 373 genomes inconsistent in total rRNA copy number between RefSeq annotation data and in silico rRNA products. Among the 373 genomes, 112 had more rRNAs defined from in silico identification and 261 had more rRNAs in RefSeq annotation. Three genomes had the same rRNA copies but showed discordance in the 16S rRNA or 23S rRNA copy number between RefSeq annotation and in silico rRNA products. There was also a genome where the 16S rRNA gene was not assembled properly at all (GCF_000281215.1; pseudogenic_rRNA). Therefore, we used 15,521 genomes for the subsequent analysis.

To identify whether the rRNA operon is linked or not, the ITS length was calculated from each rRNA operon. Using the threshold of 1500 bp in ITS length for the rRNA operon linkage status, three groups of bacteria were classified: 14,677 linked genomes, 270 mixed genomes, and 574 unlinked genomes (Figure 3B). The differences from previous studies on taxonomy and operon status are shown in Figure 3C and rRNA operon information of all genomes is available in Appendix A.

### 3.2. Correlations between Genomic Features of Symbionts and rRNA Operon Disruption

Of 15,521 complete genomes, 61 genera showed conservation of the unlinked rRNA operon for all genomes, which accounted for 88.5% (508 genomes) of the total unlinked genomes. The remainder were not conserved in each genus, with a significant reduction in rRNA copies (Figure 3D,E). Of the rRNA operon-unlinked genera, 16 genera that included 451 genomes among 574 unlinked genomes are listed in Appendix A with their genomic and phenotypic characteristics. Most of them are obligate symbiotic bacteria with insect hosts, and two genera, *Dehalococcoides* and *Thermus*, have different lifestyles from other symbiotic bacteria, such as organohalide respiration and high-temperature tolerance. Based on the SymGenDB [26], 9467 symbiotic bacteria were identified. Among 574 unlinked genomes, 413 genomes were classified as symbiotic bacteria. A phylogenetic tree of all rRNA operon-unlinked genomes showed the dominance of symbiotic bacteria and phylogenetic conservation of rRNA operon-unlinked status (Figure 4). The conservation of the rRNA operon status was common in the symbiotic genera, which contains only symbiotic species, and appear as all linked genomes or all unlinked genomes: 262/299 symbiotic genera with only linked genomes and 22/299 symbiotic genera with only unlinked genomes (Appendix A).

In symbiotic bacteria, genomic features showed significantly different distribution between linked and unlinked genomes (Figure 5). Unlinked symbionts had higher genome adenine–thymine (AT) content, lower 16S rRNA copy number, and smaller genome size compared with linked symbionts, and the comparison of five genomic features used in the PCA plot showed significant differences between linked and unlinked genomes and between mixed genomes and unlinked genomes (*p* < 0.001, Kruskal‒Wallis rank sum test and Dunn’s test). Among the genomic features of symbionts, the average of the whole genome AT contents was 64.9% in unlinked genomes, while averages of linked genomes and mixed genomes were 51.4% and 50.1%, respectively (Appendix A). Likewise, average AT contents of 16S rRNAs were also higher in unlinked symbionts than linked symbionts. AT-biased nucleotide composition, exhibited in reduced genomes of symbionts, is known to correlate to host‒symbiont co-speciation and accelerate molecular evolution [35,36]. The length of 16S and 23S rRNA was significantly different between linked symbionts and unlinked symbionts, but the correlation with rRNA operon linkage or symbiosis to the host is unclear (Appendix A).

In a comparison of the unlinked genomes of nonsymbiotic bacteria and symbiotic bacteria, genomic features also showed significant differences (Figure 6). The AT content of the whole genome and 16S rRNA was higher in unlinked symbionts than in unlinked nonsymbionts. Whole genome AT content displayed an impressive difference, with 64.9% of unlinked symbionts and 49.8% of unlinked nonsymbionts on average. Moreover, unlinked symbionts had a much shorter genome size than unlinked nonsymbionts, while the average 16S rRNA length showed no significant trend in comparison. Genome size of unlinked nonsymbionts was significantly reduced from linked symbionts (the average of genome size: 3856 Mbp (linked symbionts), 2923 Mbp (unlinked nonsymbionts); Welch’s two-sample *t*-test, *p* < 0.001) but at the same time was significantly larger than unlinked symbionts (1683 Mbp in average; Welch’s two-sample *t*-test, *p* < 0.001). The 16S rRNA copy number was slightly biased to unlinked symbionts, but the averages did not differ much from those of all unlinked genomes (the average 16S rRNA copy number: 1.538 (all unlinked genomes), 1.501 (unlinked symbionts), 1.667 (unlinked nonsymbionts); Welch’s two-sample *t*-test, *p* = 0.03886). Additionally, in unlinked conserved genera, there were no major differences in 16S and 23S rRNA length, as shown in Appendix A. However, in terms of the whole genome size and whole genome AT content, the two nonsymbiotic genera showed remarkable distribution compared to other genera (*p* < 0.00001, ANOVA and Tukey’s HSD test); *Gimesia* had the largest genome size (7901 Mbp in median) and *Thermus* had the lowest genome AT content (31.4% in median).

### 3.3. Functional Gene Alterations in rRNA Operon-Unlinked Symbionts

Representative characteristics of symbiotic bacteria include not only genomic features but also functional features such as disruption of specific metabolic pathways [36,37]. Therefore, we analyzed whether the unlinked genome has a functional gene alteration shown in symbiotic genomes. To minimize functional differences derived from a taxonomic distance, the analysis was conducted only within specific phylum. The unlinked genome tended to be functionally poor compared to the linked genome (Figure 7). When eggNOG-mapper results were counted based on the COG functional categories, the average number of linked genomes in the phyla Campylobacterota, Spirochaetota, and Chloroflexota was 1388, 1801, and 2516, respectively, except for poorly characterized ones (R and S category), whereas unlinked genomes averaged 1146, 1354, and 1058, respectively. Each functional category showed different patterns. In particular, in the case of the transcription (K) category, the unlinked genome was only 50% of the linked in all three phyla (59:32, 149:75, and 202:99 in Campylobacterota, Spirochaetota, and Chloroflexota, respectively) (paired *t*-test, *p* = 0.03). The energy production and conversion in metabolism (C) and inorganic ion transport and metabolism (P) categories also fell to 65% and 44%, respectively (paired *t*-test, *p* = 0.05; 0.03). In the remaining categories, the number of linked and unlinked was similar, or there was a larger number in unlinked.

RNase plays an important role in cell physiology and acts as a mediator in most RNA metabolisms. In particular, various RNases play major roles in rRNA degradation and maturation [38]. Therefore, we examined whether there is a relationship between the rRNA operon status and the number of RNases (Figure 8). The number of each RNase was generally conserved within the genus level. The largest difference in number was the RNase_T family (PF00929.25), with an average of 8.8 detected in *Vibrio* but none in *Dehalococcoides*. When the phylogenetic relationship was corrected, we confirmed that the unlinked genome had significantly fewer RNase_E_G (PF10150.10), RNase_T, and RNase_PH family (PF01138.22) (Brownian motion, *p* = 0.0002, 0.013, and 0.028, respectively). 

## 4. Discussion

A public genome repository may have genomes that have incorrect species names or are poorly assembled [39]. Therefore, one issue to keep in mind when performing an analysis is to check the quality of the data to prevent unreliable results. Refseq, which is a relatively more curated database than GenBank, is no exception [16]. GTDB not only carried out taxonomy re-classification but also made it possible to properly curate the genomes against misassemblies, contamination, incomplete genomes, etc. Compared to the previous study, we confirmed that the taxonomy of 696 genomes was reclassified at the genus level with GTDB, and the rRNA operon status of 53 genomes changed with a more rigorous genome screening process. The change in rRNA operon status was because the circular genome and strand of each rRNA gene were considered. For example, in the case of GCF_002197085.1, which changed from unlinked to linked, the genome size was 3,504,252 bp, the end position of 16S rRNA (+strand) was at 3,503,539 bp, and the start position of 23S rRNA (+strand) was at 437 bp. The distance between two rRNA genes is more than 3M bp, but considering the circular genome, the length of ITS is 1149 bp, which was classified as linked.

In the case of unlinked genomes that sometimes appear in the genus, it is difficult to distinguish whether real rRNA operons have all broken or assembly errors occurring in the rRNA gene family, especially in the case of a genus with only one unlinked genome. This is because unlinked genomes dramatically decreased in rRNA copy number compared to other genomes in the genus, and some genomes contained partial sequences of rRNA (in the form of “Note = 16S ribosomal RNA rRNA prediction is too short” in the annotation file). It was thought that assembly error occurred due to the presence of two or more paralogous genes [40], and only 16S rRNA and 23S rRNA of different pairs were assembled. Therefore, in the above genomes, it seems necessary to check the assembly quality through other appropriate methods.

From our results, most unlinked bacteria were symbionts and the genomes showed reduced genome size and high AT content as compared with linked symbionts. Genome reduction of symbionts is well known as a result of adaptation to hosts, and these symbionts often show the loss of genes related to DNA repair, resulting in AT-biased nucleotide composition in the bacterial population [36]. The fact that symbionts with an unlinked rRNA operon have significantly higher AT content than the linked genome indicates that obligate symbionts with close association with the host, such as bacterial endosymbionts of ticks like *Rickettsia* that are maternally inherited, show rRNA operon disruption [41]. This is also supported by the tendency for an increase in synonymous guanine–cytosine (GC) content and genomic GC content in many bacterial species [42].

Nonsymbionts with unlinked rRNA operons have a significantly larger genome size than unlinked symbionts, but a shorter genome size than linked symbionts. Among the unlinked nonsymbionts, *Thermus* and *Dehalococcoides*, which are phylogenetically conserved, have unique lifestyles. *Thermus* is a genus known as thermophilic bacteria, which lost a wide range of genes similar to obligate symbionts and showed higher levels of genomic rearrangements [43,44]. *Dehalococcoides* is known to live in the environmental sediment and to dechlorinate pollutants with remarkably small genomes [45]. Although *Thermus* showed the lowest AT content among unlinked conserved genera, the correlation between high genomic GC content and adaptation to a thermophilic environment is controversial [46,47]. Like the obligate symbionts that have a strong association with the environment and undergo genomic adaptation, these unlinked nonsymbionts are expected to undergo genomic adaptation to restricted environments, accompanied by rRNA operon disruption, but further studies are needed for detailed associations.

The regulation of the rRNA transcription process is related not only to single transcription of rRNAs but also to a rate of translation and the amount of protein synthesis [7]. rRNA regulation constitutes a stringent mechanism to respond to the surrounding nutritional contents and affect its growth rate [48,49]. Furthermore, the synthesis of the 70S ribosome is made from one 16S rRNA and one 23S rRNA. Therefore, the broken state of the rRNA operon, which has an advantage in storing regulatory information in the transcription process and helps to efficiently produce ribosomes, seems to be evidence of weak or no selective pressure on the stringent regulation of rRNA transcription.

Functionally, the most significant difference between rRNA operon-unlinked genomes and linked genomes was the reduction in the transcription (K) category, which includes several transcription regulatory genes. This has been reported in various symbiont bacteria [50,51], and researchers were able to use metabolites produced by the host without any additional control. This is a characteristic of bacteria showing a closer relationship with the host [52]. This host-dependent characteristic is also equivalent to reductions in several metabolism categories. In this context, bacteria abandon unnecessary genomic elements and express genes with a smaller dynamic range as a response to fixed environmental fluctuations. The environment may create no need for complex control in the transcription process, and less complexity of transcriptional regulation leads to a loss of transcriptional factors and metabolic pathway. These weakened selective forces for complex rRNA regulation could result in broken rRNA operon conservation through adaptive genomic rearrangement.

Meanwhile, assuming that the rRNA pair is advantageous when they are apart from each other, one candidate hypothesis can be proposed. If the co-transcriptional process cannot be maintained due to the unlinked operon, the stoichiometric ratio between 16S and 23S rRNA collapses, which may lead to an increase in the fitness in some cases. The function of 16S and 23S rRNA is protein biosynthesis by comprising 70S ribosome (30S and 50S subunits). Normally, the ribosome in an exponential phase is stable enough to be recycled [53]; however, degradation occurs under a slow-growth state or starvation conditions [54]. When degradation occurs in the subunits as well, the 30S and 50S subunits show imbalanced degradation rates [55]. Since the working form of the 70S ribosome needs to satisfy the stoichiometric ratio of 1:1, surplus subunits are generated when 16S and 23S rRNA are linked and co-transcribed. Transcription of excess DNA in nutritional deprivation could be an abuse of energy resources. Therefore, this could have acted as a hidden driving force for 16S and 23S rRNA to be expressed independently.

Additionally, oligotrophic bacteria living in the open ocean do not significantly change the number of ribosomes in the growth state using a post-transcriptional level regulatory mechanism such as riboswitch and show the same growth rate with fewer ribosomes compared to copiotrophic bacteria [56]. It can be inferred that, under oligotrophic environments, bacteria have developed survival strategies in the form of reducing what can be reduced and replacing what can be replaced. This hypothesis could be supported by studies revealing that the strength of the promoters located in the front of each rRNA is different or confirming the rate of transcription/decay of rRNA and conversion to 30S/50S subunits. Recently, with the concept of RNA velocity, it was possible to track the RNA abundance in real time [57]. Further research using this analysis is needed to see how the unlinked rRNA operon and microbial adaptation are related.

RNases E and G have a paralogue relationship with each other and tend to decrease in an unlinked genome. RNase E shortens pre-16S, and 5S rRNA is cleaved by RNase III in the rRNA maturation stage [58]. In addition, it initiates most pre-tRNA processing [59]. In the case of RNase T, which had the largest copy number variation between genera and tended to decrease in the unlinked genome, the 3′ residue of 23S rRNA in the RNase III cleavage product was shortened and was also related to tRNA processing [60]. Lastly, RNase PH catalyzes the 3′ end processing of tRNA [61]. To summarize the above, unlinked genomes tend to decrease RNases, related to the synthesis of ribosomes and tRNA for protein production.

RNases are involved in degradation as well as the maturation of RNAs. Interestingly, RNase E as the endoribonuclease and RNase PH as the exoribonuclease initiate the degradation of rRNA during starvation [62]. As described above, not only inhibiting ribosome production, but also disrupting a ribosome in starvation state positively affects survival by reducing the translation rate and providing a building block for other biosynthesis [63]. However, there are few rRNA gene copies in the unlinked genomes, leading to a decrease in RNases E and PH rather than being maintained. On the other hand, RNase III, which cleaves the precursors transcribed from the rRNA operon to initiate the processing, may be less necessary in the unlinked genome, but there was no significant difference compared to linked genomes. Since RNase does not have a single role, it is difficult to clarify its relevance to the unlinked rRNA operon. Therefore, it seems necessary to study how the rRNA processing of operon-unlinked bacteria occurs by focusing on the operation of RNases.

## 5. Conclusions

Our research suggested that unlinked bacteria were associated with symbiosis characteristics. These included genomic features such as high AT content and genome reduction, as well as functional features such as the reduction of metabolism-related genes and RNases. For symbionts, the disruption of the rRNA operon seems to be weak selection due to the decrease in the importance or efficiency of protein synthesis, or positive selection due to minimal transcription of each rRNA. Since these results were derived from an in silico study conducted based on a large-scale genome sequence, it is necessary to design experiments on operon-unlinked bacteria to verify them. We hope that these results will be an interesting topic for future evolutionary research.

## Figures and Tables

**Figure 1 biology-09-00440-f001:**
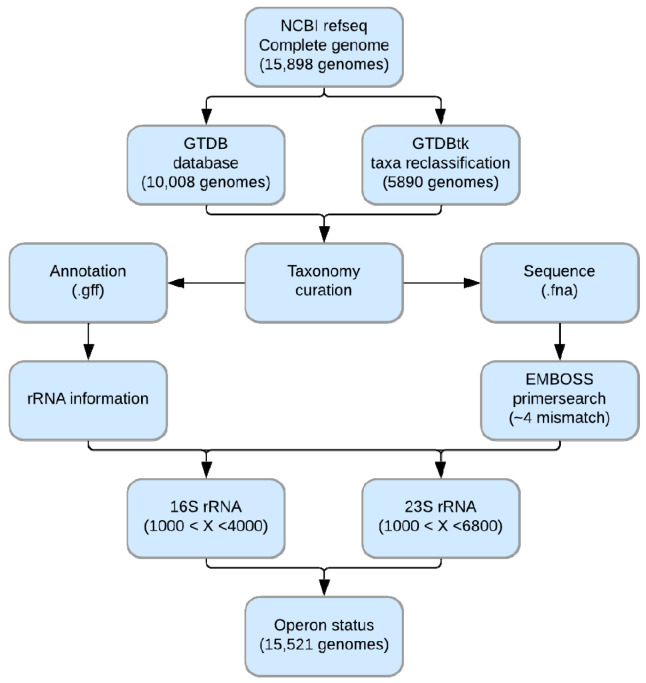
Flowchart of operon status identification from the complete genome sequence.

**Figure 2 biology-09-00440-f002:**
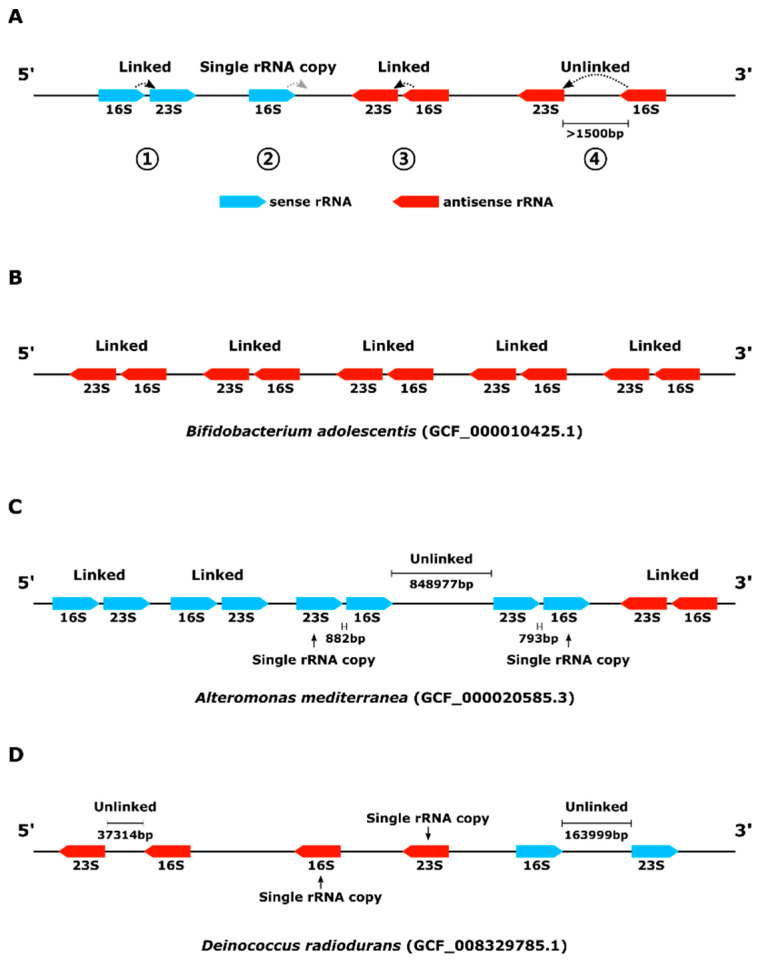
Schematic diagram of rRNA operon identification in the genome sequence. (**A**) rRNA operon diagram; (**B**) rRNA operon-linked genome; (**C**) rRNA operon-mixed genome; (**D**) rRNA operon-unlinked genome.

**Figure 3 biology-09-00440-f003:**
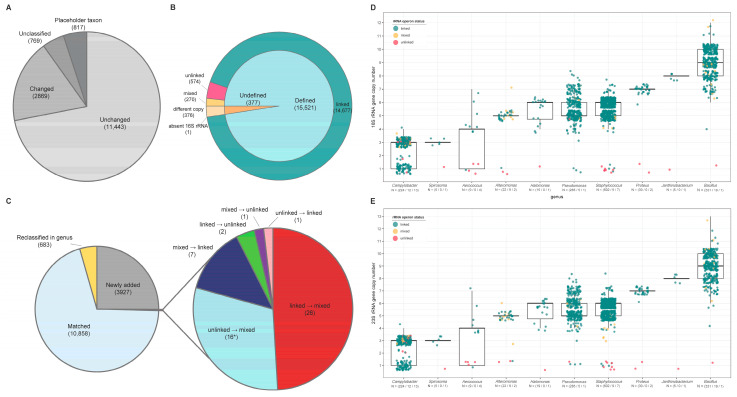
Results of genome screening. (**A**) Changes in taxonomy at the species level as a result of Genome Taxonomy Database (GTDB) reclassification; (**B**) the rRNA operon status of genomes used in this analysis; (**C**) differences from the previous analysis. Changes in taxonomy and operon status were indicated. * The group “unlinked → mixed” contained 13 “reclassified in genus.” (**D**,**E**) Distribution of rRNA gene copy numbers in a genus with a partially unlinked rRNA operon. The number of linked, mixed, and unlinked genomes for each genus was displayed on the *x*-axis, and each genome was plotted with green, yellow, and red dots. The thick line of the box represents the median, the line at the end of the box represents the first and third quartiles, and whiskers are 1.5× interquartile ranges. Dots that do not overlap whiskers are outliers. The number of 16S and 23S rRNA genes verified by in silico PCR was used. (**D**) 16S rRNA and (**E**) 23S rRNA.

**Figure 4 biology-09-00440-f004:**
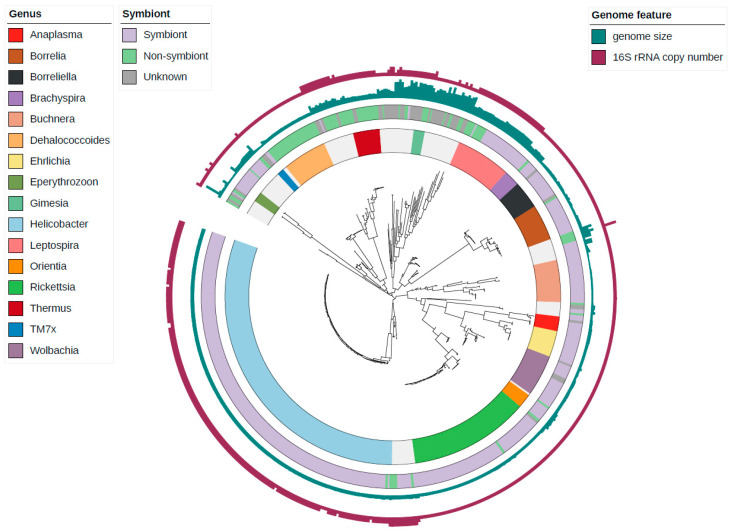
Features of each unlinked genome with phylogenetic tree. It was created in the “classify” process of GTDBtk and contained 574 unlinked genomes. The innermost strip-type circle represents clades conserved as unlinked at the genus level. The next outer circle indicates whether it was considered a symbiont of the genome, and if classification failed at the species level, it was marked as unknown. The outer two bar charts show the genome size and 16S rRNA copy number, respectively.

**Figure 5 biology-09-00440-f005:**
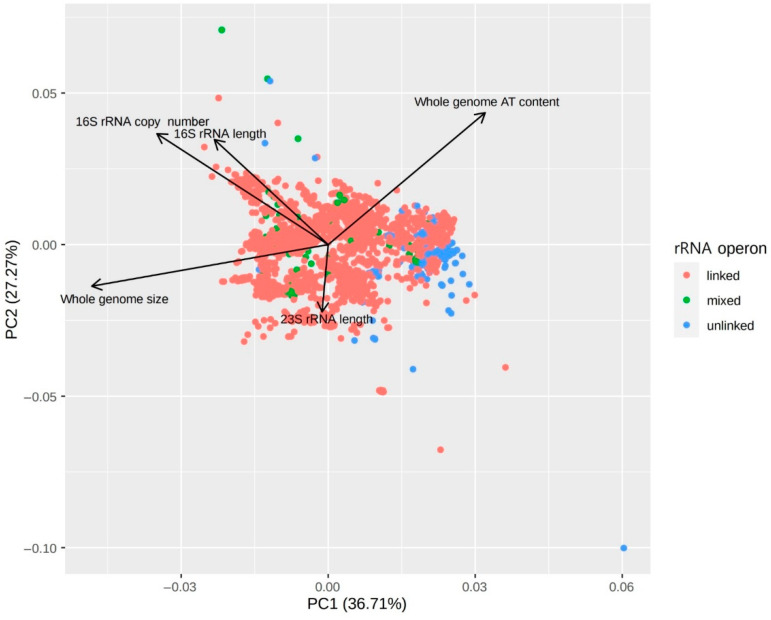
PCA plot of genomic features of symbiotic bacteria. Five genomic features showing the variation of 9467 symbiotic genomes with the rRNA operon linkage status in the principal component analysis (PCA). Each dot represents one bacterial genome. rRNA operon-unlinked bacteria were more distributed along the direction of the higher genome adenine–thymine (AT) content, lower 16S rRNA copy number, and smaller genome size. All five genomic features were significantly different between rRNA operon-linked and unlinked genomes and between mixed and unlinked genomes (*p* < 0.001, Kruskal‒Wallis rank sum test and Dunn’s test).

**Figure 6 biology-09-00440-f006:**
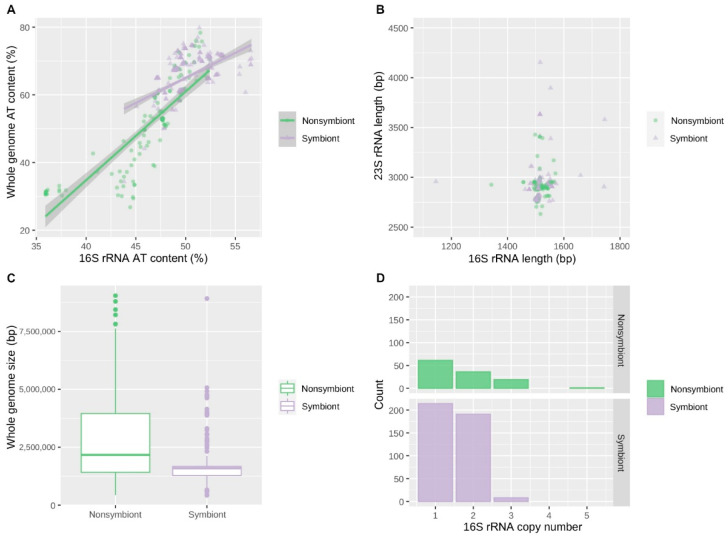
Comparisons of genomic features between symbiotic bacteria and nonsymbiotic bacteria with unlinked rRNA operons. The differences in genomic features between symbionts and nonsymbionts that have only unlinked rRNA operons are displayed. (**A**) The AT content of the whole genome sequence and the average AT content of 16S rRNAs in each bacterial genome were plotted with the linear regression and the 95% confidence intervals. The AT content of the genome and 16S rRNA was significantly different between symbionts and nonsymbionts (*p* < 0.001). (**B**) The average length of the 16S rRNAs and 23S rRNAs of each genome was plotted. (**C**) The boxplots of the whole genome size of symbionts and nonsymbionts. The whole genome size was significantly different between symbionts and nonsymbionts (the average of the whole genome size: 1.683 Mbp (symbionts), 2.923 Mbp (nonsymbionts); *p* < 0.001). (**D**) The histograms of 16S rRNA gene copies on genomes of symbionts and nonsymbionts.

**Figure 7 biology-09-00440-f007:**
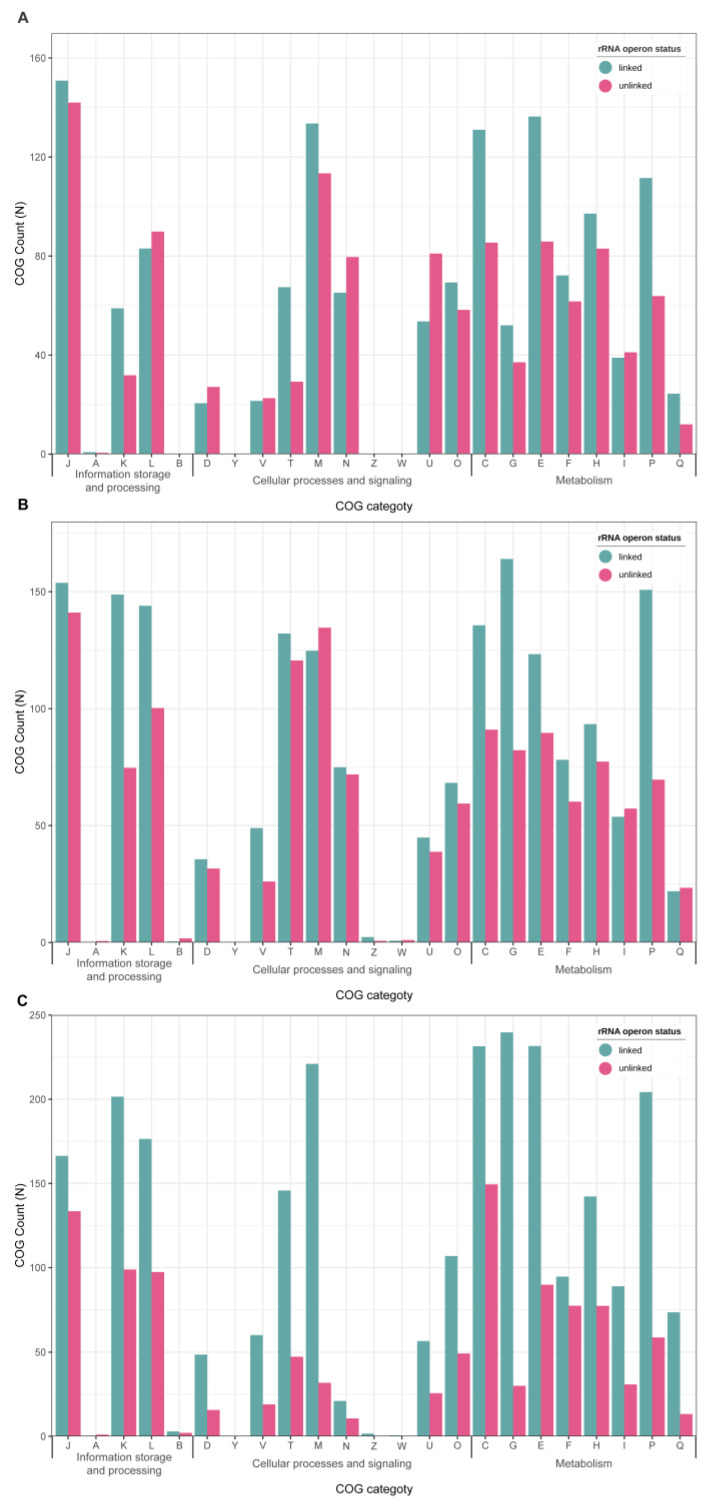
Quantitative abundance of Clusters of Orthologous (COG) categories between linked and unlinked genomes. Mixed genomes are excluded, and the average number of each group is shown. (**A**) Campylobacterota; (**B**) Spirochaetota; (**C**) Chloroflexota. COG functional categories: J, Translation, ribosomal structure, and biogenesis; A, RNA processing and modification; K, Transcription; L, Replication, recombination, and repair; B, Chromatin structure and dynamics; D, Cell cycle control, cell division, chromosome partitioning; Y, Nuclear structure; V, Defense mechanisms; T, Signal transduction mechanism; M, Cell wall/membrane/envelop biogenesis; N, Cell motility; Z, Cytoskeleton; W, Extracellular structures; U, Intracellular trafficking, secretion, and vesicular transport; O, Post-translational modification, protein turnover, chaperones; C, Energy production and conversion; G, Carbohydrate transport and metabolism; E, Amino acid transport and metabolism; F, Nucleotide transport and metabolism; H, Coenzyme transport and metabolism; I, Lipid transport and metabolism; P, Inorganic ion transport and metabolism; Q, Secondary metabolite biosynthesis, transport, and catabolism.

**Figure 8 biology-09-00440-f008:**
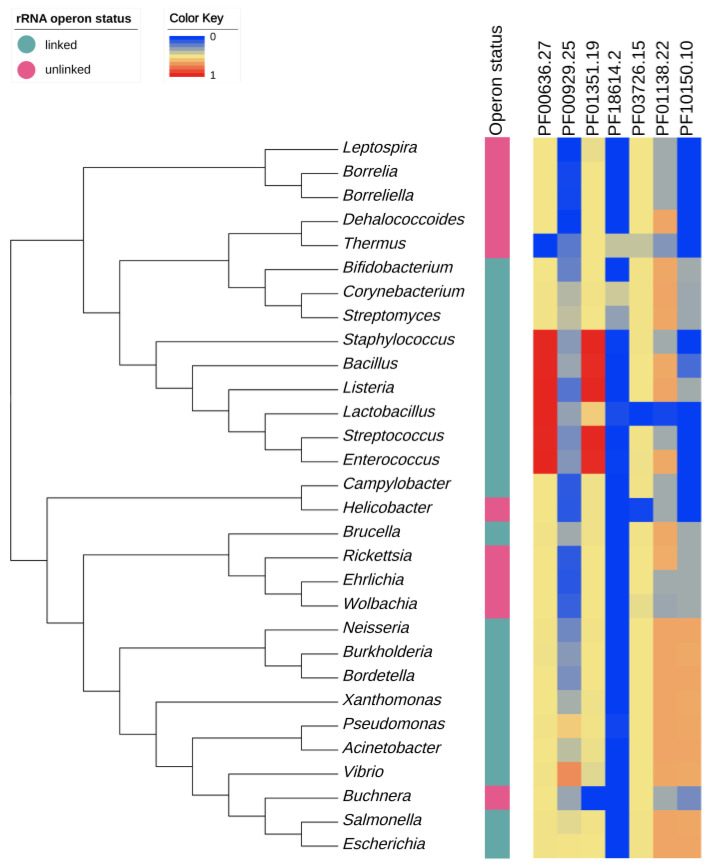
Heatmap drawn based on the number of RNases in the 30 genera. The phylogenetic relationship was determined by randomly selecting one genome for each genus and pruning it from the tree made by GTDBtk. After calculating the maximum number of each Pfam in all the genomes belonging to 30 genera, the relative amount was expressed as a heatmap. Pfam family: PF00636.27, Ribonuclease_3; PF00929.25, RNase_T; PF01351.19, RNase_HII; PF18614.2, RNase_II_C_S1; PF03726.15, PNPase; PF01138.22, RNase_PH; PF10150.10, RNase_E_G.

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
