# Peer review of "Enhanced Symbiotic Characteristics in Bacterial Genomes with the Disruption of rRNA Operon"

_biology, 2020, doi:10.3390/biology9120440_

Round 1
Reviewer 1 Report
The authors aimed to investigate about functional gene alteration in rRNA operon-unlinked genome through the Clusters of Orthologous Genes (COG) annotation. Moreover, because operon is one of the microbial genetic regulatory systems, the authors believe about the presence of certain genetic selection associated with transcription/translation regulation or other functional metabolic pathways. Their results provided basic information on what genomic changes occurred in bacterial adaptation to their environment with rRNA operon-unlinked state and could be an interesting topic for future evolutionary research.
The study is well written, is easy to follow and covers an hot topic, but some minor issues should be improved before publication.
The manuscript needs moderate English change and grammar correction.
Several typos should be corrected thorough the text.
Abstract Section: the authors should explain the aim of the study;
Conclusion Section: This paragraph required a general revision to eliminate redundant sentences and to add some "take-home message".
Reviewer 2 Report
In this study authors investigated 15,898 genomes and identified 574 unlinked rRNA genomes, which are found to be phylogenetically conserved and most of them are symbionts. Authors further studied the additional features, like AT content, functional categories, and abundance of RNase genes, which are different in unlinked rRNA operon genomes compared to linked genomes. This study is interesting in terms of understanding the abundance of unlinked genomes and its role in shaping bacterial community and evolution.
Although the article appears to hold technical and scientific soundness, the number of identified unlinked genomes is small, and authors often did not consider normalizing the genome numbers in various comparison analysis, which reduces the strength of the result.
- In line 49, authors need to introduce clearly co-transcriptional rRNA operon (linked), and transcriptionally independent rRNA operon (unlinked). Then throughout the manuscript need to present linked and unlinked rRNA operons and genomes consistently, to avoid confusion.
- To improve clarity, in fig 2C, authors should keep distance between 23S and 16S rRNA genes indicated by ‘single rRNA copy’.
- In figure 3A, 'Charged' should be Changed. Need higher resolution images of 3D and E.
- According to the figure 3, the abundance of unlinked rRNA operon is very low. Copy number reduced to near 1 for both 16S and 23S when operon is unlinked. If copy number is associated with unlinked operon, reduction tendency should also present in mixed operon. Unlinked 16S operon is missing in Pseudomonas genus. Authors need to mention and discuss the possible reasons of these phenomena.
- "Among 574 rRNA operon-unlinked genomes, 413 genomes were classified as symbiotic bacteria." How significant is this observation? What is the p value?
- "A phylogenetic tree of all rRNA operon-unlinked genomes showed the dominant proportion of symbiotic bacteria and a phylogenetic conservation of rRNA operon-unlinked status (Figure 4)". What is the conservation status among linked symbionts?
- Figure 4 needs scales of genome size and copy number.
- In figure 5, unique features of unlinked genomes are presented. Authors need to present how prominent those observations are in the two genera, Dehalococcoides and Thermus, where most of identified unlinked genomes belonged.
- "In comparison between unlinked genomes of non-symbiotic bacteria and symbiotic bacteria, genomic features also showed significant differences (Figure 6)." Did authors consider specific genera/species for this analysis?
- "Genome size of unlinked non-symbionts was significantly reduced from linked symbionts (the average of genome size: 3,856 Mbp (linked symbionts), 2,923 Mbp (unlinked non-symbionts); Welch Two Sample t-test, p = 1.717e-06)." What is the difference between linked and unlinked symbionts?
- In line 241, can author consider an analysis method that can normalize the number of genomes. This is also applicable in some other analysis of this study.
- In line 254, authors need to consider equivalent numbers of linked and unlinked genomes. Authors also need to consider whether poor annotation is the cause of low number of functional categories.
- “If the co-transcriptional process cannot be maintained due to the unlinked operon, the stoichiometric ratio between 16S and 23S rRNA collapse, which may lead to an increase of the fitness in some cases.” Did authors find different strength of the promoters located in the front of 16S and 23S rRNA?
- Did authors measure and compare the expression profile of 16S and 23S rRNA in linked and unlinked bacteria? It will be important to profile the transcriptomes of these two different types of genomes to support this study and understand the biology favoring these identified phenomena.
- If unlinked rRNA bacteria have reduced genetic capacity, they are likely to have a parasitic life-style. Can authors discuss the biology that enable these bacteria to be a member of symbiotic community?
Reviewer 3 Report
The authors have investigated the rRNA operon linkage and identified unlinked rRNA operons in a total of 574 bacterial genomes showing strong symbiotic genomic features (e.g., reduced genome size and high AT content), containing significantly fewer genes than rRNA operon-linked bacteria in ‘transcription’ and ‘energy production and conversion in metabolism’ categories, and revealing decreased RNases related to the synthesis of ribosomes and tRNA processing. The authors further concluded that disruption of rRNA operon may be one of the tendencies associated with the characteristics of bacteria requiring low dynamic range.
I believe that the authors have provided sufficient background, explained the well-established methodology clearly, presented the results appropriately, and withdrawn conclusions based on available data. The manuscript is well-written. I have no major technical concerns about this manuscript but a few minor editorial and grammatical suggestions with a few of the listed here:
Line 61, please clarify “slower growing taxa”
Line 93, please choose a different word rather than “calculated” and a couple of other cases throughout the entire manuscript because it is not appropriate to “calculate” phylogenetic relationships.
Line 357, weakened
Lines 380 and 393, RNases but not RNase.
Line 396, please re-write the phrase “Since RNase does not have a single role, …” which is confusing as it is written currently.
Please be consistent with several terms, such eggnog-mapper or “eggnog mapper”
Round 2
Reviewer 2 Report
None.